# Spinal Muscular Atrophy after Nusinersen Therapy: Improved Physiology in Pediatric Patients with No Significant Change in Urine, Serum, and Liquor 1H-NMR Metabolomes in Comparison to an Age-Matched, Healthy Cohort

**DOI:** 10.3390/metabo11040206

**Published:** 2021-03-30

**Authors:** Leon Deutsch, Damjan Osredkar, Janez Plavec, Blaž Stres

**Affiliations:** 1Department of Animal Science, Biotechnical Faculty, University of Ljubljana, SI-1000 Ljubljana, Slovenia; leon.deutsch@bf.uni-lj.si; 2Department of Pediatric Neurology, University Children’s Hospital, University Medical Centre Ljubljana, SI-1000 Ljubljana, Slovenia; damjan.osredkar@kclj.si; 3Faculty of Medicine, University of Ljubljana, SI-1000 Ljubljana, Slovenia; 4National Institute of Chemistry, NMR Center, SI-1000 Ljubljana, Slovenia; janez.plavec@ki.si; 5Department of Automation, Biocybernetics and Robotics, Jožef Stefan Institute, SI-1000 Ljubljana, Slovenia; 6Faculty of Civil and Geodetic Engineering, Institute of Sanitary Engineering, University of Ljubljana, SI-1000 Ljubljana, Slovenia; 7Department of Microbiology, University of Innsbruck, A-6020 Innsbruck, Austria

**Keywords:** spinal musular atrophy, nusinersen, 1H-NMR metabolomics, males, females, serum, liquor, urine, healthy control cohort

## Abstract

Spinal muscular atrophy (SMA) is a genetically heterogeneous group of rare neuromuscular diseases and was until recently the most common genetic cause of death in children. The effects of 2-month nusinersen therapy on urine, serum, and liquor 1H-NMR metabolomes in SMA males and females were not explored yet, especially not in comparison to the urine 1H-NMR metabolomes of matching male and female cohorts. In this prospective, single-centered study, urine, serum, and liquor samples were collected from 25 male and female pediatric patients with SMA before and after 2 months of nusinersen therapy and urine samples from a matching healthy cohort (*n* = 125). Nusinersen intrathecal application was the first therapy for the treatment of SMA by the Food and Drug Administration (FDA) and the European Medicines Agency (EMA). Metabolomes were analyzed using targeted metabolomics utilizing 600 MHz 1H-NMR, parametric and nonparametric multivariate statistical analyses, machine learning, and modeling. Medical assessment before and after nusinersen therapy showed significant improvements of movement, posture, and strength according to various medical tests. No significant differences were found in metabolomes before and after nusinersen therapy in urine, serum, and liquor samples using an ensemble of statistical and machine learning approaches. In comparison to a healthy cohort, 1H-NMR metabolomes of SMA patients contained a reduced number and concentration of urine metabolites and differed significantly between males and females as well. Significantly larger data scatter was observed for SMA patients in comparison to matched healthy controls. Machine learning confirmed urinary creatinine as the most significant, distinguishing SMA patients from the healthy cohort. The positive effects of nusinersen therapy clearly preceded or took place devoid of significant rearrangements in the 1H-NMR metabolomic makeup of serum, urine, and liquor. Urine creatinine was successful at distinguishing SMA patients from the matched healthy cohort, which is a simple systemic novelty linking creatinine and SMA to the physiology of inactivity and diabetes, and it facilitates the monitoring of SMA disease in pediatric patients through non-invasive urine collection.

## 1. Introduction

The 5q spinal muscular atrophy (SMA) is a rare neuromuscular disorder, which leads to progressive atrophy and weakening of skeletal muscles due to the progressive loss of motor neurons [1]. It is characterized by degeneration of the motor neurons in the anterior horn of the spinal cord, resulting in atrophy and weakness of the voluntary muscles of the limbs and trunk. With the incidence of about one in 11,000 live births, it was, until the development of disease-modifying drugs, the most common genetic cause of child deaths [2]. The gene for the survival motor neuron (SMN) was localized in 1990 by two different groups [3,4] on chromosome 5q13 and in 1996 was identified as the disease-causing gene. The SMN protein is produced by two genes, the telomeric SMN1 gene, which is deleted or interrupted in patients with SMA, and the centromeric SMN2 gene, which differs from SMN1 by five nucleotides and is present in several copies (Appendix A). SMN1 produces full-length transcripts, while SMN2 in 90% produces transcripts without exon 7, because of the C to T mutation that produces an exon-splicing suppressor sequence [5,6]. In SMA patients, SMN2 is present in at least one copy and is usually truncated because of C to T substitution (c.840C → T), since in 90–95% of the cases, the exon is spliced out of the produced SMN. The remaining 10% produce the SMN protein in full length, which indicates a low SMN expression [7]. SMN is a 38 kDa protein that is expressed in all somatic tissues and is located in the nucleus, in the cytoplasm, and in the axons of motor neurons. Its role is not yet fully understood, but the phenotype of the SMA depends largely on the number of SMN2 gene copies present [8].

However, SMA is a motor neuron disease that also affects the skeletal muscle, heart, kidney, liver, pancreas, spleen, bone, connective tissues, and immune systems [9]. 

Novel therapies for the treatment of SMA have emerged that modify the natural course of the disease by modifying the expression or replacing mutated genes involved in the development of SMA. Such treatment options are currently nusinersen [10] (an antisense oligonucleotide that modifies mRNA splicing) (Appendix A), onasemnogene abeparvovec [11] (gene replacement therapy), and risdiplam [12] (a small molecule that modifies mRNA splicing). Nusinersen was the first drug approved for SMA. It was approved by the US Food and Drug Administration (FDA) in December 2016 and by the European Medicines Agency (EMA) in June 2017 [13,14]. Nusinersen (Appendix A) is an antisense oligonucleotide that promotes the inclusion of exon 7 into mRNA transcripts of SMN2. It binds to an intronic splice site in intron 7 of SMN2 and inhibits the action of other splicing factors, thereby promoting exon 7 incorporation into the mRNA. This leads to the production of a fully functional SMN protein. Antisense oligonucleotides do not cross the blood–brain barrier, which means that they must be administered intrathecally [15,16,17]. The nusinersen drug improved motor function and increased the amplitude of muscle action potential of the ulnar and peroneal nerve. An autopsy analysis showed the uptake of nusinersen into motor neurons throughout the spinal cord and into neurons in the brainstem [18]. Increased motor function was manifested as an increased ability to sit or walk independently [19,20,21], increased bite force [22], and increased hand strength [23]. Although nusinersen is administered intrathecally, which requires a lot of expertise, it has been shown that such application is well tolerated and safe [24]. Nusinersen is available in many countries for most types of SMA patients, depending mostly on inclusion criteria and financing defined by the country of residence. However, there is an enormous need for real-world evidence of nusinersen efficacy, for better understanding of the variability of effect and side effects in a broader cohort of SMA patients [25].

In the last two decades, new approaches have been developed in the natural sciences. These include various ‘omics techniques: from genomics, which is an important method for establishing the SMA diagnosis, to metatranscriptomics, proteomics, and metabolomics to monitor and investigate disease [7]. A systemic approach to therapy during early development is most likely to maximize the positive clinical outcome. Metabolomics is a useful method to evaluate the metabolites that we can identify in different biological samples, leading to the end of a cascade of biological processes, hence helping us to understand the molecular phenotype and the underlying metabolic mechanisms [26]. Nuclear magnetic resonance spectroscopy (NMR) is one of the most widely used approaches due to its minimal sample preparation, non-destructive measurement method, quantitative aspect, and high reproducibility [27,28,29]. 1H-NMR has been used to study the modulation of metabolites to cellular stress [30], breast cancer markers [31], acute pancreatitis [32], influence of metabolome on health and disease [33], biomarkers for Crohn’s disease and ulcerative colitis [34], obesity [35], coronary heart disease and stroke [36], next to physiological deconditioning through inactivity and hypoxia [37]. In line with these reports, serum creatinine was only recently identified as a potential biomarker for monitoring the SMA progression of denervation with decreasing levels reflecting the severity of the disease [38]. However, it is currently unknown whether (i) creatinine concentration showed improvement upon early medical intervention with nusinersen therapy, (ii) whether these changes are detectable in liquor, serum, or urine samples using 1H-NMR metabolomics approaches, and (iii) in relation to age- and sex- matched healthy male and female cohorts, next to (iv) whether additional biomarkers can be identified using urine samples relative to healthy cohorts. 

In this study, the effects of nusinersen on the 1H-NMR metabolomes of three bodily fluids urine, liquor, or serum were explored in male and female pediatric patients with SMA (Appendix A). Matched healthy male and female cohorts (Appendix A) were used to explore urine as the more ubiquitous and accessible matrix for SMA detection and monitoring. Ensemble multivariate statistical approaches (nonparametric and parametric) coupled to machine learning were used to interrogate the metabolomics data in order to establish the significance of differences between SMA before and after treatment and a healthy cohort in order to build the respective sample classification model based on the most important and validated urine biomarkers for the first time.

## 2. Results and Discussion

### 2.1. Comparison of 1H-NMR Metabolomes of Urine, Serum, and Liquor Samples before and after Nusinersen Intervention: Positive Effects of Nusinersen Therapy Clearly Preceded or Took Place Devoid of Significant Rearrangements in the Metabolomic Makeup of Serum, Urine, or Liquor

SMA samples of urine, serum, and liquor were collected before and after the 4th application of nusinersen therapy and processed as described below. Medical checkup before the first and after the 4th treatment showed significant improvements at the level of better movement, easier writing and sitting or standing, and feeling more strength next to easier finger extension, which were all measured according to The Children’s Hospital of Philadelphia Infant Test of Neuromuscular Disorders (CHOP INTEND) [39], Hammersmith Functional Motor Scale (HFMS), or Expanded Hammersmith Functional Motor Scale (HFMSE) [40] scales or Motor Function Measurement (MFM) [41] tests (Table 1). Twenty patients showed improvement in moving, alongside 8 in head control, 7 in eating, 3 in breathing, 5 in wheelchair control, 10 in tiredness, 4 in hygiene, 5 in mood, 7 in speech, 2 in sleep, and one in excretion after the 4th application of nusinersen. 

In contrast, the nonparametric tests (npMANOVA; *p* > 0.05; FDR corrected) used in our analyses showed no significant differences in metabolites in samples collected before and after the 4th application of nusinersen, irrespective of the sample matrix (urine, serum, liquor), sex, or data transformation and normalization procedures. These findings were further corroborated by additional parametric analyses using analyses as implemented within Metaboanalyst (PLSDA, random forest) (Figure 1) and extensive modelling using Just Add Data Bio (JADBIO), all showing no significant difference between the metabolomes collected before and after 4th application of nusinersen. Lastly, the exploration of statistical power also supported the same conclusion. The differences between the metabolic profiles of samples collected before and after the 4th application of nusinersen were so small that at least a two orders of magnitude larger sample size per group (amounting to thousands of samples) would be needed to detect significant differences between the two SMA groups at the level of metabolites and their concentrations obtained by 1H-NMR.

Taken together, these results show that the positive effects of nusinersen therapy (Table 1) preceded or took place devoid of significant changes in the actual metabolomics makeup of all three matrices: serum, urine, or liquor (Figure 1). Therefore, the existence of subtle differences could be explored only using approaches utilizing detection thresholds beyond the routine 1H-NMR approaches, such as mass spectrometry methods with higher sensitivity and the possible identification of metabolites present at nM concentrations. However, the relative ease of sample preparation, the ability to quantify metabolite levels, the high level of experimental reproducibility, and the inherently nondestructive nature of NMR spectroscopy support the selection of 1H-NMR as the preferred platform for clinical metabolomic studies [42].

The congruency between the results obtained using various bodily fluids (urine, serum, liquor) showed that the lack of differences in the metabolite profile due to nusinersen therapy were reproducible on all three matrices and hence indeed remarkably small. Our approach also introduced urine as a more straightforward sampling approach to monitor the overall physical status in SMA patients relative to a healthy population compared to more complex serum or liquor samples utilized so far.

### 2.2. Search for Additional SMA Biomarkers Utilizing Routine 1H-NMR in Urine: Sex Differences, Lower Overall Metabolite Concentrations and Diversity, and Creatinine Content

The collection of urine samples from SMA patients and also from the healthy cohort enabled us to compare the two groups in search of biomarkers for the delineation of SMA and healthy controls next to the detailed targeted 1H-NMR metabolite biomarker search. The ensemble statistical approaches (npMANOVA, MetaboAnalyst; JADBIO) applied to the data matrices (all analyzed metabolites *x* all samples) clearly identified the existence of significant differences between male and female metabolism on one side next to differences between the healthy cohort and SMA on the other.

First, npMANOVA showed the importance of sex (F = 54.9; *p* = 0.0001) and SMA status before or after treatment (F = 20.7; *p* = 0.0001) as significant, while their interaction or SMA status itself (pre vs. post) were not, irrespective of the three different approaches to data preparation and transformation (composition (%), Box–Cox, Log(*x* + 1)). 

Second, the same distinction between the groups of metabolites detectable in urine samples was obtained using PLSDA and randomForest classification as implemented in MetaboAnalyst approach (Figure 2, Figure 3, Appendix A), clearly showing that the differences between male and female physiology were significant at the level of overall urine data encompassing SMA and healthy cohort (*p* < 0.05). Using the same data sizes and matched composition, the differences between males and females were still observable in the healthy cohort (Figure 3) (*p* < 0.05), while they were less pronounced in the SMA dataset due to the larger scatter observed in SMA groups of males and females (*p* < 0.85).

We further explored differences in measured concentrations and numbers of different metabolites between SMA and the healthy cohort. The cumulative concentration of metabolites found in the urine of SMA patients corresponded to 58% of those found in the healthy cohort, which was a significant reduction (*p* < 0.05). The higher concentrations of metabolites observed in males relative to females were not significant due to the twice as large scatter observed in SMA groups of males and females (*p* < 0.05). These results show the general picture of a significantly lower overall concentration of urine metabolites in SMA patients relative to the healthy cohort. In addition, the presence/absence pattern of metabolites, i.e., the number of detected metabolites was also significantly higher (*p* < 0.001) in the healthy cohort relative to SMA; however, it was not significantly different between SMA male and female patients, again, due to the large scatter observed in SMA groups of males and females, relative to that observed in matching healthy cohorts of males and females (Appendix A). In comparison, the healthy male cohort exhibited a significantly larger number of metabolties (*n* = 178.2 ± 10.8) in comparison to the healthy female cohort (*n* = 154.9 ± 24) used in this study, providing future guidance for introducing female participants to such experiments. This is the first report describing the existence of such differences in the context of SMA; hence, the underlying mechanisms for the observed significant differences in metabolic makeup between SMA and healthy next to males and females warrant further analyses in the future. 

Some tentative parallels exist with research conducted in the fields of exercise and inactivity. In the context of exercise medicine, analyses of exercise showed that runners experienced a profound systemic shift in blood metabolites related to energy production (especially from the lipid super pathway) and that following the 3-day exercise period, significant 2-fold or higher increases in 75 metabolites persisted for longer than a day [43]. A recent review on exercise metabolic changes revealed that in total, at least 196 metabolites changed their concentration significantly within 24 h after exercise in at least studies, signifying the importance of daily exercise bouts for the maintenance of metabolic diversity and concentration [44]. On the other hand, in the context of body deconditioning due to physical inactivity, similar findings as observed here for the SMA cohort were reported in controlled bed-rest studies. Three-week inactivity resulted in a 30% reduction in the number of statistically significantly connected metabolites, a 2.5 times reduction in the number of interactions, and diminished metabolic diversity within the human body. Conversely, the short-term complete inactivity exhibited also rather similar physiological changes (Appendix A) such as insulin resistance, bone and muscle resorption, constipation, changes in lipid metabolism, and progressively negative interactions with microbiome [29,45,46] that are all listed as part of the SMA makeup as well.

Third, extensive statistical analyses using MetaboAnalyst (Appendix A) and machine learning JADBIO modeling (Figure 4) were adopted to explore the importance of metabolites measured in urine samples. In total, 60,340 models were trained based on the complete urine metabolomics dataset comprising the SMA before, SMA after, and Healthy participants. PLSDA based on all three groups showed that the only significant difference existed again between SMA on one side and the healthy cohort on the other (Figure 4A). The most interpretable model was identified as the ridge logistic regression with the penalty hyperparameter lambda equal to 0.1 with an area under the curve (AUC) value of 0.958 (Appendix A). In addition to AUC, all other threshold metrics were also statistically significantly different from baseline. Data were preprocessed with constant removal and standardized. Features were selected based on Test-Budgeted Statistically Equivalent Signature (SES) algorithm with hyperparameters: maxK = 2, alpha = 0.01, and budget = 3 *nvars. The performance of the model when only creatinine from urine is used is 97.378% (with 95% CI from 94.043% to 100%). Out of all metabolites, creatinine measured in urine samples was the most significant metabolite uniformly separating the healthy group from the SMA group (Figure 4B and Appendix A), whereas no additional metabolic features could be identified to separate SMA before from SMA after, in analogy with the other tests described in this study (Appendix A). We used the trained model on the test part of our data (30% of our total dataset) and achieved the validation performance with an AUC of 0.970. Our work shows that decreasing creatinine concentrations in urine can be used as an additional easy way to measure differentiations between SMA patients and healthy groups (Figure 5 and Appendix A) in analogy with creatinine in serum, which was only recently put forward as a potential biomarker for monitoring the SMA progression of denervation with decreasing levels, reflecting the severity of the disease [38]. However, in response to the open question put forward of whether creatinine responded to molecular therapies [38], the results presented in this study showed that the levels of creatinine did not change significantly in response to the application of nusinersen therapy. Inclusion and analysis of the urinary 1H-NMR metabolomics data following extended nusinersen therapy is projected to further answer this question in the future.

Spinal muscular atrophy patient data suggest that spinal muscular atrophy is a disease affecting neurons, which has diverse consequences for multiple tissues (skeletal muscle, heart, kidney, liver, pancreas, spleen, bone, connective tissues, intestinal tract, and immune systems) [9]. Space-exploration studies adopting bed-rest, e.g., our PlanHab experiment [29,37,45,46], represent a controlled environment for elucidation of the effects physical inactivity as such. Valuable insight was obtained into body deconditioning including insulin resistance, bone and muscle resorption, constipation, mood changes such as depression, negative changes in lipid metabolism, and inflammatory interactions with the microbiome and cardiovascular hypertension (Appendix A) [29,45,46], next to modifications in bacterial metabolism and mucosal turnover in the gut, contributing to the transfer of inflammatory compounds into the bloodstream [47]. 

These similarities point to a complex interplay and joint effects of physical inactivity and the congenital SMA disease of the patients. We highlight the fact that these are all distinct medical conditions which, despite their different etiologies, share certain characteristics of their metabolic phenotype and clinical characteristics that deserve further exploration.

Decreased creatinine concentrations were observed in urine samples of bed-rest immobilized healthy male participants of the planetary habitat exploration studies [37] (Appendix A). The reintroduction of exercise effectively alleviated and completely reversed the negative effects observed in the PlanHab project [29,37,45,46]. In another study, bed-rest immobilized participants that received vibrational therapy [48] showed numerous benefits relative to controls and could be considered as a step in the physical activation of SMA patients after nusinersen therapy, similar to the prevention and treatment of many diseases, including diabetes and obesity in the future. 

## 3. Materials and Methods

### 3.1. Patients and Healthy Volunteers

This was a single-center study. Biological samples were collected from all patients, clinically diagnosed, and genetically confirmed with SMA. Patients were younger than 19 years and were treated with nusinersen at the Department of Child, Adolescent and Developmental Neurology at the University Children’s Hospital Ljubljana, Slovenia. 

In March 2017, nusinersen was available through the early access program. Five children received the application before was approved by the European Medicines Agency (EMA). After the approval, The Health Insurance Institute of Slovenia offered the treatment to all children and eligible young adults. Those who decided on treatment were enrolled in the study between March 2017 and by June 2020 and received four consecutive applications of nusinersen. 

The study and all experimental protocols were approved by the National Medical Ethics Committee of Republic of Slovenia (0120-305/2018/6 and 0120-305/2018/11). All participants and/or their legal guardians signed the informed consent. The study was registered at ClinicalTrials.gov (accessed on 29 January 2021) under the identifier NCT04587492.

In total (Appendix A), 48 samples of urine (25 before treatment and 23 after the 4th application of nusinersen), 41 samples of serum (21 before and 22 after), and 46 samples of liquor (24 before and 22 after) were collected by medical professionals at the University Children’s Hospital Ljubljana, Slovenia. The SMA cohort consisted of 15 female patients (age: 8.8 ± 5.5 years; height: 131 ± 21 cm; weight: 26 ± 15 kg) and 10 male patients (age: 9.3 ± 5.1 years; height: 131 ± 20 cm; weight: 28 ± 16 kg). The matching healthy female and male cohort consisted 48 female volunteers (age: 9.4 ± 3.5 years; height: 136 ± 20 cm; weight: 32 ± 13 kg) and 77 healthy male volunteers (age: 9.6 ± 4.3 years; height: 142 ± 24 cm; weight: 36 ± 16 kg). Daily urine samples were collected for three consecutive days to capture daily variation in routines and dietary habits. All samples (SMA and the healthy group) were included in a newly established Slovenian Urine NMR database (manuscript in preparation).

### 3.2. Evaluation

We analyzed the number of SMN2 copies in all genetically confirmed SMA patients. Evaluation of patients were performed before the initiation of treatment, before the 5th application (after 6 months). Neurological, pulmological, gastroenterological, endocrinological, and psychological exams were performed by pediatric specialists before the start of treatment. The physical capabilities were performed by standardized test by a physiotherapist: The Children’s Hospital of Philadelphia Infant Test of Neuromuscular Disorders (CHOP INTEND) [39], Hammersmith Functional Motor Scale (HFMS), or Expanded Hammersmith Functional Motor Scale (HFMSE) [40] scales or Motor Function Measurement (MFM) [41] test. Testing was performed by a physiotherapist trained to use the tests, depending on the age and capabilities of the patient, before the treatment and at all follow-up examinations. The same test was used for all time-points in each particular patient.

### 3.3. Treatment with Nusinersen

All patients were treated with intrathecal (IT) application of nusinersen. The application was performed under controlled hospital environment. The majority of applications were performed without sedation or under sedation with midazolam or with a combination of midazolam and ketamine. The IT application in very anxious patients was performed under general anesthesia. All patients were recommended to be well hydrated for at least one day before application. 

The IT injection of nusinersen was performed on days 0, 14, 30, and 60 in all patients in a standard dose of 5 mL (12 mg/mL). In smaller children, the dose was appropriately reduced after the application; all children were advised to lie prone for 2 h to reduce the risk of post lumbar puncture (LP) symptoms. To reduce the risk of post lumbar puncture symptoms, all patients were advised to lie prone for at least 2 h after IT application and were monitored for potential side effects for the entire duration of the study. 

After the first application, patients were dismissed from the hospital one day after the application and on the same day in the evening for the following applications.

### 3.4. Sample Collection

For all patients treated with nusinersen who received the medicine intrathecally, samples of urine, blood serum, and cerebrospinal fluid (CSF) were collected before the initiation of treatment, and after the 4th (2 months) application of nusinersen. All obtained samples were frozen at −20 °C for further analysis. Replicate stability analyses were performed as described before [29,37] (Appendix A).

### 3.5. Metabolome Analysis Using Proton Nuclear Magnetic Resonance (^1^H-NMR)

All collected samples were centrifuged (1.5 mL) at 10,000× *g* for 30 min to remove fine particles. Then, 400 µL of supernatant were mixed with 200 µL ^1^H-NMR buffer as described before [49] and stored at −25 °C until analysis. Serum samples were filtered using 3 kDa colons (Amicom Ultra 3 kDa (Merck Millipore, Burlington, MA, USA)) for the additional removal of large molecules [50]. Before analysis, samples were thawed at room temperature and transferred into a 5 mm NMR tube. TSP was used as an internal standard for quantification, as described before [49] 

An Agilent Technologies DD2 600 MHz spectrometer equipped with a 5 mm HCN Cold probe was used for the acquisition of NMR spectra at 25 °C. The ^1^H NMR spectra of the samples were recorded with a spectral width of 9.0 kHz, relaxation delay of 2.0 s, 32 scans, and 32 K data points. A double-pulsed field gradient spin echo (DPFGSE) pulse sequence was used for water suppression. Total correlated spectrum (TOCSY) was measured with ^1^H spectral widths of 7.0 kHz, 4096 complex points, a relaxation delay of 1.5 s, 32 transients, and 144 time increments. An exponential and cosine-squared function were used for apodization. Zeros were filled before Fourier transform. VNMRJ (Agilent/Varian) software was used for processing urine and liquor NMR spectra. 

Serum spectra were acquired on the same spectrometer equipped with a 24-sample automation system processed with Topspin v. 4.0.9 software (Bruker, Billerica, MA, USA).

Metabolites were identified with the support of the Chenomx Compound Library extended to the Human Metabolome Database [51,52], giving access to chemical shift profiles of 674 compounds used in analyses. The number of database-derived chemical shift profiles of metabolites used in analyses was further decreased by the procedures described below.

### 3.6. Bioinformatics and Statistical Analysis

The resulting spectra were consequently analyzed using targeted quantitative metabolomics using Chenomx NMR Suite version 8.6 (Chenomx, Inc., Edmonton, AB, Canada). For the latter, all spectra were randomly ordered for spectral fitting using the ChenomX profiler. An ensemble approach to data analysis was utilized, employing three different approaches to asymmetric sparse matrix data analysis, establishing significant differences between tested groups as follows: nonparametric MANOVA (PERMANOVA) [53], MetaboAnalyst [54,55], and JADBIO [56]. 

First, for npMANOVA, each compound concentration obtained was analyzed as described before in three different ways [29,37]: (i) by dividing the measured concentration by the concentration of all metabolites in that sample; (ii) Box–Cox; or (iii) log(*x* + 1) transformed. The significance of metabolic differences between various groups of samples was tested using ANOSIM, NP-MANOVA, and expressed as an overlap in non-metric multidimensional scaling (nm-MDS) trait space (using Euclidean distance measures). The stress function was used to select the dimensionality reduction, whereas Shepard’s plots were used to describe the correspondence between the target and obtained ranks [57]. Benjamini–Hochberg significance correction for multiple comparisons was used as described before [45,46].

Second, for MetaboAnalyst, log or cube root transformation in connection to Mean or Pareto scaling was utilized as implemented in MetaboAnalyst [54,55] followed by supervised classification using partial least squares discriminant analysis (PLSDA) method, random forest (RF), and pathway enrichment analysis. PLSDA results were cross-validated with a caret package implemented in MeatboAnalyst. The most important metabolites identified by PLSDA were determined according to variable importance in projection (VIP). The randomForest package implemented in MetaboAnalyst was used for supervised classification between different groups of interest. The most important features defined by RF were ranked by a mean decrease in classification accuracy. Hierarchical clustering was performed according to the VIP scores to obtain a heat map representing differences in metabolic profiles between samples and groups. Euclidean distance, Pearson’s correlation, and Spearman’s correlation were used as similarity measures and Ward’s linkage was used as a clustering algorithm. Statistical power for the identification of significant differences before and after treatment was also calculated using MetaboAnalyst Statistical Power module. MetaboAnalyst and ggplot2 were used for graph generating. 

KEGG human pathway libraries were used for metabolic pathway and enrichment analysis. For topological analysis, the globaltest analysis method and relative betweenness centrality were used. Significant pathways were determined using the raw *p*-value, Holm–Bonferroni *p*-adjusted value, and adjusted *p*-value using the False Discovery Rate. The impact of pathways was calculated using pathway topology analysis. 

Metabolite Set Enrichment (MSEA) was used to identify biologically significant patterns between quantitative metabolome data from different groups. HMDB compound names were used to link to the KEGG database. Enrichment analysis was performed using the globaltest package implemented in MetaboAnalyst. The enrichment ratio was calculated by dividing observed hits and expected hits.

Last, Just Add Data Bio (JADBIO), a web-based auto machine learning platform for analyzing potential biomarkers [56], was used. The JADBIO platform was designed for predictive modeling and to provide high-quality predictive models for diagnostics using state-of-the-art statistical and machine learning methods. Personal analytic biases and methodological statistical errors were eliminated from the analysis by the autonomous exploration of various settings in modeling steps producing more convincing discovered features to discriminate between SMA and the healthy group. JADBIO 1.1.164 with extensive tuning effort and 6 CPU was used to model various dataset selections next to the overall 336 metabolites observed in urine samples in all groups (healthy versus SMA group) by splitting the total urine metabolite data into a training set and a test set in a 70:30 ratio. The training set was used for model training and the test set was used for model evaluation.

The resulting model can be obtained as part of Appendix A (ESM2) and run with java executor for the classification of novel urine samples based on 1H-NMR metabolomes in further exploration.

## Figures and Tables

**Figure 1 metabolites-11-00206-f001:**
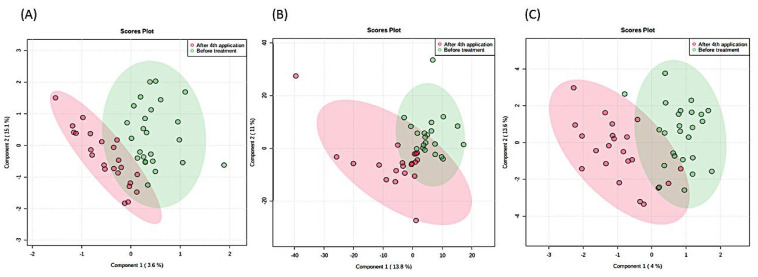
A Partial least square discriminant analysis (PLSDA) of 1H-NMR metabolomes of SMA patients based on (**A**) urine, (**B**) serum, and (**C**) liquor before (red) and after (green) the 4th application of nusinersen. Ellipses designate 95% confidence intervals for each group. The differences are not significant (ensemble statistical approach (npMANOVA, MetaboAnalyst, JADBio)).

**Figure 2 metabolites-11-00206-f002:**
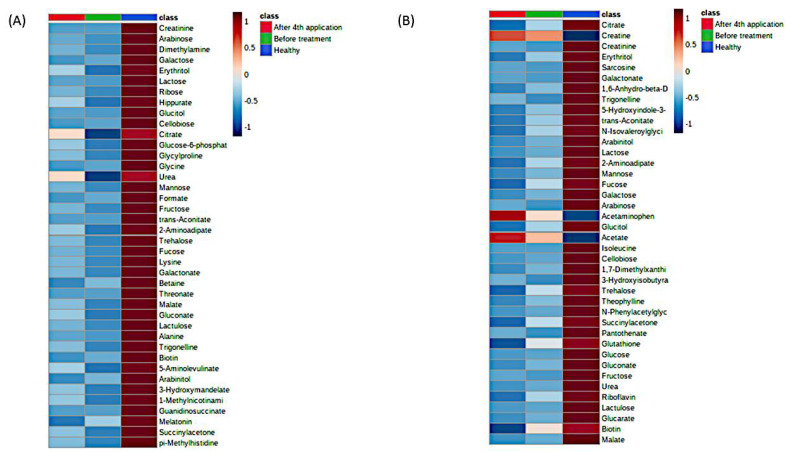
MetaboAnalyst heatmaps of urine 1H-NMR metabolomes representing existing differences between SMA cohort (before and after) and healthy cohort, for males (**A**) and females (**B**). The differences on metabolic level between the before-treatment (green) and after 4th application (red) groups were not significant due to the large variability in SMA samples. The most differentiating metabolites were selected by the PLSDA variable importance in projection (VIP) score, where decreasing metabolites were presented with negative values (blue color) and increasing metabolites were presented with positive values (red color). Individual data are presented in the Appendix A.

**Figure 3 metabolites-11-00206-f003:**
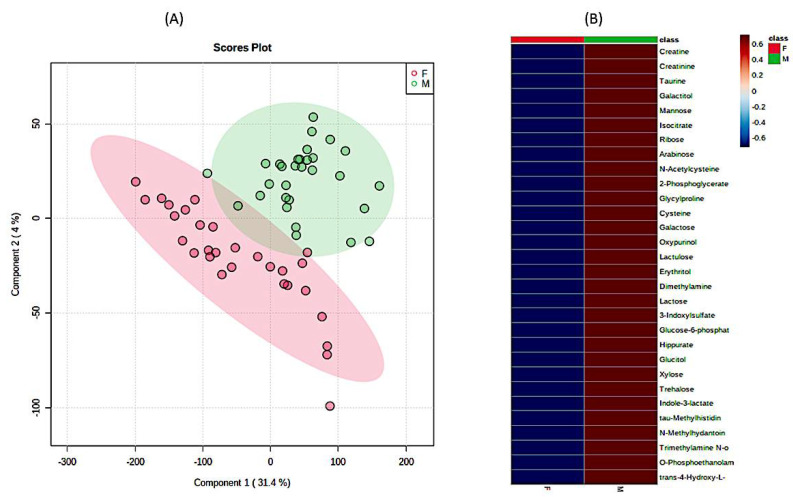
The significant differences in urine 1H-NMR metabolomic profiles between females and males in matched healthy cohorts. (**A**) PLSDA analysis and (**B**) heatmap analysis representing existing differences in the first 30 most important metabolites. (F—females (red), M—males (green)). The most differentiating metabolites were selected by the PLSDA VIP score, where decreasing metabolites had negative values (blue color) and increasing metabolites had positive values (red color).

**Figure 4 metabolites-11-00206-f004:**
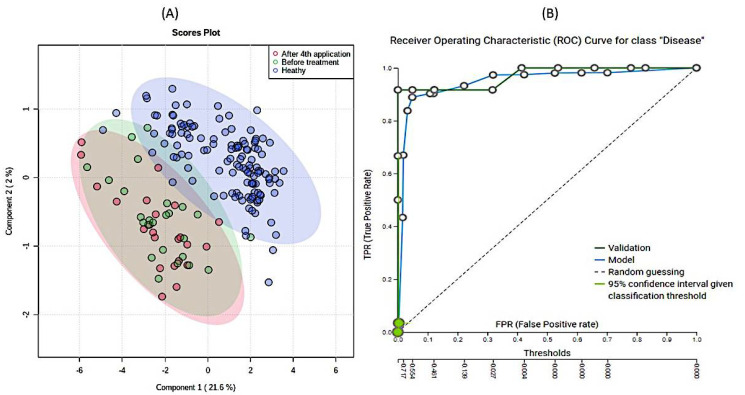
Results of partial least square discriminant analysis (PLSDA) (**A**) and (**B**) Receiver Operating Characteristics curve of modeling of the data. A PLSDA-based ordination of 1H-NMR urine metabolomes: healthy (blue), before (red), and after (green) 4th application of nusinersen. Ellipses designate 95% confidence intervals for each group (**A**). (**B**) Receiver Operating Characteristic (ROC) curve for SMA patients obtained with model. The Just Add Data Bio (JADBIO) model is available as part of the electronic Appendix A for the classification of novel 1H-NMR metabolomic data.

**Figure 5 metabolites-11-00206-f005:**
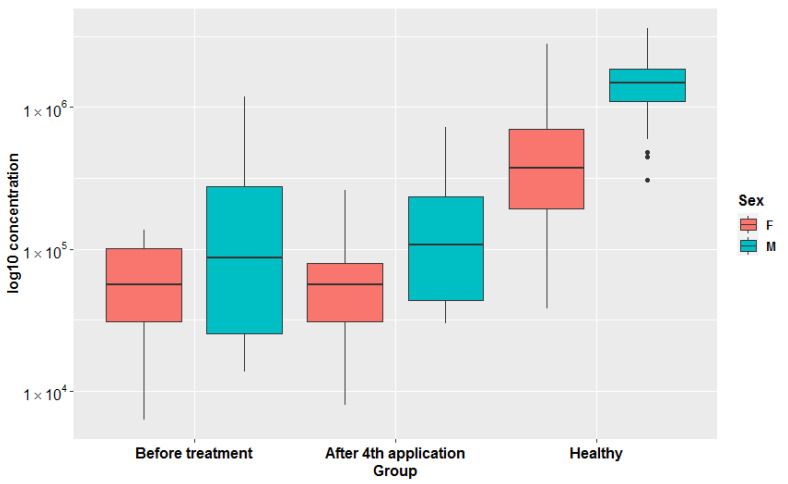
Box-plots representing log10 transformed creatinine concentrations in all three groups (before treatment, after 4th application, healthy) in females and males separately. Original concentrations are presented in Appendix A.

**Table 1 metabolites-11-00206-t001:** Clinical data obtained from spinal muscular atrophy (SMA) patients. Due to the small cohort size, patients’ data were anonymized.

Patient	Sex	SMA Type	SMN Copies	Age at 1st App	Weight at 1st App	Height at 1st App	Summary Score at 1st App	Ambulatory at 1st App	Ambulatory at 7th App	Ambulatory Change	Other
1	1	2	3	13.5	57	152	7.5	0	0	No	More strength in the legs while laying on the back
2	1	2	4	5.8	16	103	30	0			More strength
3	2	3	3	4.3	12.4	92	87.5	1	1	No	More strength
4	2	2	3	11.7	42	134	7.5	0	0	No	She stands easier when supported when going to the toilet
5	2	2	3	1.7	10	78	45.3	0	0	No	Stronger
6	1	3	4	7.6	37	136	100	1	1	No	Walks easier
7	2	2	4	12.3	22.5	135	54.7	0	0	No	Writes easier
8	2	2	3	8.6	22.5	126	15	0	0	No	Sits easier and better torso control
9	2	2	4	18.8	50	143	15	0	0	No	Talks easier and moves upper and lower limbs easier
10	2	3	3	13.8	44	160	100	1	1	No	Muscle pain after long walk
11	2	2	3	11.3	13	125	15.6	0	0	No	Stronger voice
12	1	2	3	11.6	33	144	7.8	0	0	No	More easily extends fingers
13	1	2	3	15.4	19	150	6.3	0	0	No	No changes observed
14	2	3	3	5.2	16.5	105	52.5	0	0	No	Easier movement
15	2	2	4	2.3	10.2	82	55	0	0	No	More strength
16	2	1	4	1.3	12.3	74.5	31.3	0	0	No	
17	1	2	3	6.4	13	114	17.2	0	0	No	Better movement
18	2	3	3	18.6	44	154	100	1			
19	2	2	3	1	7.6	75	59.4	0	0	No	Movement better
20	1	2	3	9.8	39.5	146	20.3	0	0	No	More strength by physiotherapists
21	1	3	4	14.2	58	174	97.5	1	1	No	Can walk further
22	2	2	4	5.9	14	110	30	0	0	No	Better movement
23	2	2	3	3.3	24	102	47.5	0	0	No	No difference
24	1	3	3	13	41	152	85	0	0	No	Better movement
25	1			9.1	30.4	138.5					
26	1			1.41	10						
27	1			3.92	45.2	152					

## Data Availability

Liquor, serum and urine matabolomic data are made available upon request to the corresponding author. Model with test data for clustering samples according to metabolite profiles are included in the Appendix A. Instructions for running a model on local machine are included in the electronic Appendix A.

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
