# Peer review of "Spinal Muscular Atrophy after Nusinersen Therapy: Improved Physiology in Pediatric Patients with No Significant Change in Urine, Serum, and Liquor 1H-NMR Metabolomes in Comparison to an Age-Matched, Healthy Cohort"

_metabolites, 2021, doi:10.3390/metabo11040206_

Round 1
Reviewer 1 Report
General comment:
Deutsch et al. investigated the effect of nusinersen treatment in children with SMA on the metabolome of various tissues and compared it to age-matched, healthy controls. They found that while clinical parameters improved, this was not reflected in changes of the metabolome of SMA patients. While the metabolome of the SMA patients did not change significantly, the authors were able to find differences between urinary creatinine concentrations in SMA patients compared to healthy controls. Finally, the SMA data is compared to adults after a period of bed rest and the authors conclude by highlighting the overlap between the conditions.
The authors are commended for providing insights into the effects of a relatively novel drug for a clinical population in urgent need for treatment as well as an improved understanding of their disease. In essence, this study could warrant a publication in the targeted journal. However, in its current form the manuscript displays major flaws which would have to be addressed adequately before it can be endorsed by this reviewer.
Revisions:
Title
Line 1: Change “Spinal muscle atrophy” to “Spinal muscular atrophy”.
Line 2-3: The title should reflect that the investigated population were children.
Line 3: The authors are strongly encouraged to leave “and persistence of inactivity phenotype” out of the title to avoid confusion for the reader.
Line 5: Add “to an age-matched, healthy control cohort”.
Abstract
Line 25: Emphasize that the investigated population were children.
Line 25-26: Correct to “were collected from [...], and urine samples from matching healthy controls [...].”.
Line 28: Add comma after “analyses”, delete “next to”.
Line 30: Add “and” after “posture”.
Line 32: Add an “a” before “cohort” or change to “cohorts”.
Line 33: Add an “a” before “reduced”.
Line 35-39: Please revise these sentences for redundancy. At least two sentences almost identically mention how urinary creatinine distinguishes SMA patients from controls.
Line 37: Change to “devoid of” instead of “devoid”.
Line 39-40: Similar to the title, the authors are encouraged to avoid overstating the connection between SMA, inactivity and diabetes. There is not enough evidence provided in this study to highlight this as a major finding.
Introduction
Line 66-68: SMA is purely a motor neuron disease. You could change the sentence to “SMA is a motor neuron disease that also affects [...].”, but the current phrasing is misleading.
Line 96: Change to “to monitor and investigate disease” or delete “and its research”.
Line 115: Emphasize that the study investigated children.
Line 117: Please correct to “An ensemble multivariate statistical approach” or “Ensemble multivariate statistical approaches”.
Line 119: This makes it appear as though the modeling approach was strictly used to “diagnose” or “detect” SMA. Which is not reflected in the manuscript. Please clarify.
Line 119-120: The mentioning of power analyses belongs in the methods section.
Line 120: Please change “Last” to “Lastly”.
Line 120-122: Consider rephrasing to “Lastly, potential similarities between the metabolome of [...] and [...] were explored.
Line 123-125: This unnecessarily overstates the implications of this study and the authors are encouraged to simply end their introduction with the previous sentence.
Results and discussion
In general, the results and discussion section does not adequately connect the text to the figures. At no place does the text directly mention the figures and it is unclear which data and graphs the authors refer to when they discuss them in the text.
Line 130: Change to “devoid of”.
Line 132-138: The authors are strongly encouraged to include any clinical data pertaining to the patient populations and the CHOP, HFMSE and MFE tests in the main manuscript. Either as a separate figure or part of what is currently figure 1. The patient data also needs to include the different degrees of severity the SMA patients have been diagnosed with. This data is vital to the study and the interpretation of the other results.
Line 146: Change to “Lastly”.
Line 154-156: Explain in greater detail why other technological platforms such as GC-MS or LC-MS could allow for a more thorough insight. This is a key limitation of the study at hand and needs to be discussed openly.
Line 160: Change to “to monitor” instead of “to monitoring”.
Line 160-161: “physical” and “body” are redundant here. Change to “physical status”.
Line 161: Add “a” before “healthy population” or change to “healthy populations”.
Line 161: Delete the comma after “population”, delete “next to” and change to “compared to”.
Line 163-165: Please include all those data (see also line 132-138). The clinical changes are mentioned in the entire manuscript, starting in the title, and continuing in abstract, introduction and the results. They are a cornerstone of this manuscript and the data needs to be included and adequately discussed.
Line 165-167: This is a conclusion not warranted based on the data provided. The contribution of less activity to the metabolome and disease course of SMA patients is purely speculative. The activity of the SMA patients was not measured in this study and the question whether changes to the metabolome (or lack thereof) reflect inactivity or the congenital disease of the patients is complex. Furthermore, it is inappropriate to treat the decreased activity of SMA patients as equal to complete bed rest. While the comparison between the metabolome and bed rest is interesting and can be part of the manuscript, the authors are encouraged to address this subject more carefully, clearly communicate its speculative nature and stay clear from overstating their findings.
Line 172-181: Please revise for clarity and in the light of the comment above. The authors mention twice how surprising their results are (line 172, line 176), immediately before stating that the results are not surprising at all (line 177). Again, the authors are encouraged to err on the side of caution when interpreting their results. Instead of mixing immobilization, bed rest, loss of gravity and SMA, they could highlight the fact that these are all distinct medical conditions which, despite their different etiologies, share certain characteristics of their metabolic phenotype.
Line 182-187: In addition to the PCA, this figure (or a separate table) should include a list or heatmap of the most prevalent metabolites quantified before the treatment and after the 4th application. Currently it is not clear to the reader which metabolites the authors measured and what the quantities were. This should also be reflected in the figure/table descriptions. Furthermore, the authors should consider performing a simple linear regression analysis between the clinical change (CHOP, HFMSE and MFE) and the changes to the metabolome (either as a whole or with individual metabolites).
Line 188: Change to “in urine” and “sex differences, lower metabolite concentrations- and diversity, and creatinine content”.
Line 192-198: As mentioned above, it is unclear what data the authors refer to as there is no connection between the text and the figures. Please clarify which figures you are referring to and make sure that all the data mentioned in the manuscript is also displayed in the figures. Additionally, the authors are encouraged to avoid putting any of their data into the supplement. Currently there are only two figures in the manuscript that display original data, there is no need for a supplement.
Line 195: In the entirety of the manuscript, replace “gender” by “sex”.
Line 195: Does “SMA status” refer to the clinical diagnosis of SMA (type 1, 2, 3 and 4) or the time point at which the patients were measured (before treatment/after 4th application)?
Line 199-202: Where is the data that clearly shows the sex differences? Make sure to include it in your figures.
Line 202-205: See comment above.
Line 206-230: See comment above. Make sure the figures display any data that is mentioned in the text, and make sure the text accurately links them together.
Line 231-249: See comment above. Assuming this part of the results & discussion section mainly surrounds figure 2, this figure is currently not adequately displaying any of the data analysis discussed in the text.
Line 249-255: Please ensure the figure contains all the data discussed in the main text and the description sufficiently informs the reader about its content. Similar to the recommendation regarding figure 1, consider including the individual values in a table or heatmap (sorted by effect size for the most prominent changes, for example). This will allow the reader to get a more accurate picture of the results, especially since the PCA turned out too crude to distinguish the changes to the metabolome of the SMA patients before and after treatment.
Line 256-257: Again, the creatinine values for the patients should be directly displayed in any of the figures, when discussed in such detail in the text.
Line 258: Delete “surprisingly”, as you previously mentioned yourself that it is not surprising you found a certain overlap in the symptoms of SMA and bed rest patients. Additionally, change “physiological symptoms” to “clinical symptoms” and make sure the symptoms are listed in figure 1 together with the other information about the SMA patients (see comment on line 132-138).
Line 259-268: Please clarify which symptoms are common between the pediatric patients with SMA and the adult patients who participated in the PlanHab study. Please be concise about it and maintain a focus on SMA.
Line 268-279: Please revise this part of the manuscript to become more precise and less speculative. It is conceivable that there is a certain overlap between SMA and the PlanHab study, which may or may not be related to the lack of physical activity. However, it is important to highlight the differences between the etiologies and make sure the results are not overstated. Consider performing a simple linear regression analysis, investigating the metabolome or certain metabolites in SMA, healthy controls and bed rest patients of the PlanHab study. If this turns out to yield data that supports the hypothesis that there is an overlap between clinical symptoms and the metabolome of the various populations, then there would be a slightly firmer ground to place the speculations on. However, even if you are able to find significant correlations between the metabolome or certain metabolites, these results should not be overstated.
Line 279-282: Consider replacing this figure with any of the original data mentioned above. Since this schematic illustration is primarily based on speculation, it currently weakens the manuscript by distracting from valid, original findings made.
Line 284-298: The concepts that physical inactivity contributes to the clinical phenotype seen in SMA patients and that vibration therapy could help alleviate some of that are interesting and conceivable. However, the study at hand does not establish a convincing direct link between those variables and this should reflect in the language and extent to which these ideas are discussed. If the authors choose to continue to include this part of the discussion in the manuscript, they are encouraged to provide data analysis associated with it (see previous comment).
Methods
Line 370-443: The authors mention that they performed “targeted metabolomics”. To the reviewers understanding, this comprises the quantification of a defined number of metabolites in a sample. It is unclear to the reader on which basis the authors quantified the metabolite concentrations and/or what they were normalized to.
Line 392: Change to “Bioinformatics and statistical analysis”.
Author Response
Response to Reviewer 1 Comments
Deutsch et al. investigated the effect of nusinersen treatment in children with SMA on the metabolome of various tissues and compared it to age-matched, healthy controls. They found that while clinical parameters improved, this was not reflected in changes of the metabolome of SMA patients. While the metabolome of the SMA patients did not change significantly, the authors were able to find differences between urinary creatinine concentrations in SMA patients compared to healthy controls. Finally, the SMA data is compared to adults after a period of bed rest and the authors conclude by highlighting the overlap between the conditions.
The authors are commended for providing insights into the effects of a relatively novel drug for a clinical population in urgent need for treatment as well as an improved understanding of their disease. In essence, this study could warrant a publication in the targeted journal. However, in its current form the manuscript displays major flaws which would have to be addressed adequately before it can be endorsed by this reviewer.
Revisions:
Title
A: The title was adapted as suggested by the reviewer.
Line 1: Change “Spinal muscle atrophy” to “Spinal muscular atrophy”.
A: Changed to spinal muscular atrophy.
Line 2-3: The title should reflect that the investigated population were children.
A: We added in pediatric patients.
Line 3: The authors are strongly encouraged to leave “and persistence of inactivity phenotype” out of the title to avoid confusion for the reader.
A: We erased “and persistence of inactivity phenotype”.
Line 5: Add “to an age-matched, healthy control cohort”.
A: We added “to an age-matched, healthy control cohort”.
Abstract
Line 25: Emphasize that the investigated population were children.
A: We added “pediatric”.
Line 25-26: Correct to “were collected from [...], and urine samples from matching healthy controls [...].”.
A: We corrected this sentence to “were collected from 25 male and female pediatric patients with SMA before and after 2 months of nusinersen theraphy, and urine samples from matching healthy cohort (n=125).”
Line 28: Add comma after “analyses”, delete “next to”.
A: Comma was added and “next to” deleted.
Line 30: Add “and” after “posture”.
A: “and” after “posture” added.
Line 32: Add an “a” before “cohort” or change to “cohorts”.
A: We added “a”.
Line 33: Add an “a” before “reduced”.
A: We added “a”.
Line 35-39: Please revise these sentences for redundancy. At least two sentences almost identically mention how urinary creatinine distinguishes SMA patients from controls.
A: Rephrased.
Line 37: Change to “devoid of” instead of “devoid”.
A: We changed to “devoid of”.
Line 39-40: Similar to the title, the authors are encouraged to avoid overstating the connection between SMA, inactivity and diabetes. There is not enough evidence provided in this study to highlight this as a major finding.
A: Rephrased as requested.
Introduction
Line 66-68: SMA is purely a motor neuron disease. You could change the sentence to “SMA is a motor neuron disease that also affects [...].”, but the current phrasing is misleading.
A: We changed the sentence to “However, SMA is a motor neuron disease that also affects skeletal muscle, heart, kidney, liver, pancreas, spleen, bone, connective tissues, and immune systems”
Line 96: Change to “to monitor and investigate disease” or delete “and its research”.
A: Changed to “to monitor and investigate disease.”
Line 115: Emphasize that the study investigated children.
A: We added “pediatric” to patients.
Line 117: Please correct to “An ensemble multivariate statistical approach” or “Ensemble multivariate statistical approaches”.
A: We corrected to “Ensemble multivariate statistical approaches.”
Line 119: This makes it appear as though the modeling approach was strictly used to “diagnose” or “detect” SMA. Which is not reflected in the manuscript. Please clarify.
A: All metabolites were included in analyses to identify those that could discern between SMA before, SMA after and Healthy cohorts.
Text was rephrased for clarity.
Line 119-120: The mentioning of power analyses belongs in the methods section.
A: We moved sentence about power analysis to the section 3.6.
Line 120: Please change “Last” to “Lastly”.
A: We changed “Last” to “Lastly”.
Line 120-122: Consider rephrasing to “Lastly, potential similarities between the metabolome of [...] and [...] were explored.
A: We rephrased this sentence to “Lastly, potential similarities between the metabolome of SMA male cohort and inactive male cohort 1H-NMR urine metabolomes from our past space exploration bed-rest studies [37] were explored.”
Line 123-125: This unnecessarily overstates the implications of this study and the authors are encouraged to simply end their introduction with the previous sentence.
A: We deleted the last sentence.
Results and discussion
In general, the results and discussion section does not adequately connect the text to the figures. At no place does the text directly mention the figures and it is unclear which data and graphs the authors refer to when they discuss them in the text.
Line 130: Change to “devoid of”.
A: Changed to “devoid of”.
Line 132-138: The authors are strongly encouraged to include any clinical data pertaining to the patient populations and the CHOP, HFMSE and MFE tests in the main manuscript. Either as a separate figure or part of what is currently figure 1. The patient data also needs to include the different degrees of severity the SMA patients have been diagnosed with. This data is vital to the study and the interpretation of the other results.
A: A novel Table 1 was constructed and is now part of the revised manuscript as requested.
Line 146: Change to “Lastly”.
A: Changed to “Lastly”.
Line 154-156: Explain in greater detail why other technological platforms such as GC-MS or LC-MS could allow for a more thorough insight. This is a key limitation of the study at hand and needs to be discussed openly.
A: We explained differences between MS and NMR methods.
Line 160: Change to “to monitor” instead of “to monitoring”.
A: Changed to “to monitor”.
Line 160-161: “physical” and “body” are redundant here. Change to “physical status”.
A: Changed to physical status.
Line 161: Add “a” before “healthy population” or change to “healthy populations”.
A: We added “a” before “healthy population”.
Line 161: Delete the comma after “population”, delete “next to” and change to “compared to”.
A: We deleted comma after “population”, deleted “next to” and changed to “compared to”.
Line 163-165: Please include all those data (see also line 132-138). The clinical changes are mentioned in the entire manuscript, starting in the title, and continuing in abstract, introduction and the results. They are a cornerstone of this manuscript and the data needs to be included and adequately discussed.
A: A novel Table 1 was constructed and is now part of the revised manuscript as requested.
Line 165-167: This is a conclusion not warranted based on the data provided. The contribution of less activity to the metabolome and disease course of SMA patients is purely speculative. The activity of the SMA patients was not measured in this study and the question whether changes to the metabolome (or lack thereof) reflect inactivity or the congenital disease of the patients is complex. Furthermore, it is inappropriate to treat the decreased activity of SMA patients as equal to complete bed rest. While the comparison between the metabolome and bed rest is interesting and can be part of the manuscript, the authors are encouraged to address this subject more carefully, clearly communicate its speculative nature and stay clear from overstating their findings.
A: Lifestyle of participants was not significantly changed or modified due to the genetic therapy. This is now part of the revised manuscript. Novel Table1 was prepared that contains this information.
In our submitted manuscript the decreased activity of SMA patients was not treated as equal to complete bed rest. The physiological parameters of both groups were contrasted and the previously unreported matching was mentioned in this paragraph.
This paragraph was rephrased to address this subject more carefully and to clearly communicate its speculative nature.
Line 172-181: Please revise for clarity and in the light of the comment above. The authors mention twice how surprising their results are (line 172, line 176), immediately before stating that the results are not surprising at all (line 177). Again, the authors are encouraged to err on the side of caution when interpreting their results. Instead of mixing immobilization, bed rest, loss of gravity and SMA, they could highlight the fact that these are all distinct medical conditions which, despite their different etiologies, share certain characteristics of their metabolic phenotype.
A: Rephrased as requested. Please see also our comment above.
Line 182-187: In addition to the PCA, this figure (or a separate table) should include a list or heatmap of the most prevalent metabolites quantified before the treatment and after the 4th application. Currently it is not clear to the reader which metabolites the authors measured and what the quantities were. This should also be reflected in the figure/table descriptions. Furthermore, the authors should consider performing a simple linear regression analysis between the clinical change (CHOP, HFMSE and MFE) and the changes to the metabolome (either as a whole or with individual metabolites).
A: A novel Figure2 was prepared in the form of heatmap of the most prevalent metabolites quantified before the treatment and after the 4th application as requested.
Line 188: Change to “in urine” and “sex differences, lower metabolite concentrations- and diversity, and creatinine content”.
We changed that subheading to “2.2 Search for additional SMA biomarkers utilizing routine 1H-NMR in urine: sex differences, lower metabolite concentrations- and diversity, and creatinine content”
Line 192-198: As mentioned above, it is unclear what data the authors refer to as there is no connection between the text and the figures. Please clarify which figures you are referring to and make sure that all the data mentioned in the manuscript is also displayed in the figures. Additionally, the authors are encouraged to avoid putting any of their data into the supplement. Currently there are only two figures in the manuscript that display original data, there is no need for a supplement.
A: Additional Table 1, new Figure 2 and new Figure 3 were prepared and are now part of the revised manuscript.
The links between the text and figures were established.
For statistical tests there is no other way to present the results than to prepare the summary of tests in the text.
Line 195: In the entirety of the manuscript, replace “gender” by “sex”.
A: “Gender” was replaced by “sex”.
Line 195: Does “SMA status” refer to the clinical diagnosis of SMA (type 1, 2, 3 and 4) or the time point at which the patients were measured (before treatment/after 4th application)?
A: We added “before and after treatment”.
Line 199-202: Where is the data that clearly shows the sex differences? Make sure to include it in your figures.
A: Statistical tests using npMANOVA, MetaboAmnalyst and JADBio were performed on data matrices and were reported in the text of our submitted manuscript. This text was amended for clarity.
We added a novel Figure2, Figure3 and Figure5.
Line 202-205: See comment above.
Line 206-230: See comment above. Make sure the figures display any data that is mentioned in the text, and make sure the text accurately links them together.
A: Novel figures were prepared and are now part of the revised manuscript. Text was rephrased for clarity as requested.
Line 231-249: See comment above. Assuming this part of the results & discussion section mainly surrounds figure 2, this figure is currently not adequately displaying any of the data analysis discussed in the text.
A: Novel figures were prepared and are now part of the revised manuscript. Text was rephrased for clarity as requested. The text was rephrased to make explicitly clear that after extensive machine learning and statistical analyses there were no other metabolites significantly differing between SMA and matched healthy cohorts that could be used in separation of the two groups except creatinine. Other metabolites were considered however, their signal in separating the groups was not significant in this study.
Line 249-255: Please ensure the figure contains all the data discussed in the main text and the description sufficiently informs the reader about its content. Similar to the recommendation regarding figure 1, consider including the individual values in a table or heatmap (sorted by effect size for the most prominent changes, for example). This will allow the reader to get a more accurate picture of the results, especially since the PCA turned out too crude to distinguish the changes to the metabolome of the SMA patients before and after treatment.
A: Partial Least Square Discriminant Analysis (PLSDA) was conducted as written in the Materials and methods and the results were only visualized in the form of PCA ordination. More than 60,000 models were tested in order to obtain the most interpretative model, within which the creatinine was the most informative compound to distinguish between the SMA and healthy matched cohort.
We have to disagree, as in order to present additional metabolites as significant in our revised manuscript the analyses would have to be performed in a way that models would be overfitting. The ensemble statistical approach was adopted precisely to rigorously report on features that contain significant signal to avoid overfitting next to minimizing false positive results.
We rephrased the text to make these observations clear.
Line 256-257: Again, the creatinine values for the patients should be directly displayed in any of the figures, when discussed in such detail in the text.
A: A novel Figure5 was prepared and is now part of the revised manuscript.
Line 258: Delete “surprisingly”, as you previously mentioned yourself that it is not surprising you found a certain overlap in the symptoms of SMA and bed rest patients. Additionally, change “physiological symptoms” to “clinical symptoms” and make sure the symptoms are listed in figure 1 together with the other information about the SMA patients (see comment on line 132-138).
A: Rephrased as requested.
Additional supplementary figure was presented reproduced with permission under CC-BY license to showcase the matching between the symptoms of the two conditions.
Also, SMA related symptoms were already listed in the submitted manuscript
Line 259-268: Please clarify which symptoms are common between the pediatric patients with SMA and the adult patients who participated in the PlanHab study. Please be concise about it and maintain a focus on SMA.
A: The matching of the symptoms with the focus on SMA was already presented in the submitted manuscript on lines 168-182.
The text was additionally rephrased as requested.
Additional two supplementary figures were introduced in Electronic Supplementary Materials, reproduced with permission under CC-BY license from our past publication, to showcase and clarify the matching between the symptoms of the two conditions.
Line 268-279: Please revise this part of the manuscript to become more precise and less speculative. It is conceivable that there is a certain overlap between SMA and the PlanHab study, which may or may not be related to the lack of physical activity. However, it is important to highlight the differences between the etiologies and make sure the results are not overstated. Consider performing a simple linear regression analysis, investigating the metabolome or certain metabolites in SMA, healthy controls and bed rest patients of the PlanHab study. If this turns out to yield data that supports the hypothesis that there is an overlap between clinical symptoms and the metabolome of the various populations, then there would be a slightly firmer ground to place the speculations on. However, even if you are able to find significant correlations between the metabolome or certain metabolites, these results should not be overstated.
A: The text was significantly revised as requested and the Figure 3 of the submitted manuscript was removed.
Line 279-282: Consider replacing this figure with any of the original data mentioned above. Since this schematic illustration is primarily based on speculation, it currently weakens the manuscript by distracting from valid, original findings made.
A: We dully considered this point and provided novel Figure2, Figure3, Figure5 plus two additional Supplementary material Figures to support this hypothesis.
We rephrased and shortened the text according to the suggestions and removed the Figure 3 of our submitted manuscript.
Line 284-298: The concepts that physical inactivity contributes to the clinical phenotype seen in SMA patients and that vibration therapy could help alleviate some of that are interesting and conceivable. However, the study at hand does not establish a convincing direct link between those variables and this should reflect in the language and extent to which these ideas are discussed. If the authors choose to continue to include this part of the discussion in the manuscript, they are encouraged to provide data analysis associated with it (see previous comment).
A: Please see also our response above. This section was rephrased, modified and shortened as requested.
Methods
Line 370-443: The authors mention that they performed “targeted metabolomics”. To the reviewers understanding, this comprises the quantification of a defined number of metabolites in a sample. It is unclear to the reader on which basis the authors quantified the metabolite concentrations and/or what they were normalized to.
Line 392: Change to “Bioinformatics and statistical analysis”.
A: Correct. Targeted metabolomics was performed as described in Materials and methods.
The quantification was done based on internal standard TSP as described in Materials and methods section.
Changed to “Bioinformatics and statistical analysis.”

Reviewer 2 Report
The purpose of this study was to determine potential metabolites that relate to spinal muscular atrophy (SMA) and how nusinersen therapy alters the metabolite profile. To do this, 25 SMA patients (15 females, 10 males) were examined before and after 2-months of nusinersen therapy. Urine, serum, and liquor samples were collected for metabolome profiling by 1H-NMR. Urine samples were compared to 125 healthy controls (48 females, 77 males). Despite apparent phenotypic improvements, nusinersen therapy did not alter the metabolome profile in SMA patients. In comparison to healthy controls, SMA patients presented reduced number and concentration of metabolites, with machine learning confirming creatinine levels distinguish SMA patients from healthy controls. These findings suggest the improvements provoked by nusinersen therapy occur before, or devoid of changes in the metabolome profile. While the use of metabolome profiling has strength to identify molecular mediators of SMA, as well as potential mediators of nusinersen therapy, there was various issues with the overall manuscript to preclude it from being accepted in its current form. Comments/suggestions to improve the manuscript can be found below.
Major
- The use of metabolome profiling has great strength to identify cellular mediators of diseased conditions. However, beyond the creatinine findings, no additional metabolites were discussed. Were other metabolites differentially expressed between the SMA and healthy controls?
- In the second to last paragraph in the Introduction, it appears that the authors specifically wanted to examine the role of creatinine with SMA, based on previous work. With a potentially large dataset, specifically looking at one metabolite minimizes the importance of looking at the whole metabolite profile. While creatinine may have been the metabolite that distinguished SMA the most, other metabolites may be involved and are worth presenting.
- In addition to group differences, relationships between phenotypic characteristics can be used to identify potential cellular mediators of physical function measures (i.e., movement, posture, and strength tests described in the abstract). It would be beneficial to run a correlation matrix between metabolites and physical function measures to see which metabolites are associated with physical derangements with SMA, and which individual metabolites may be involved in physical improvements provoked by nusinersen therapy.
- Overall, a rearrangement of the findings will greatly improve the quality of the manuscript. Specifically, first presenting how SMA alters the metabolite profile in comparison to healthy controls will show strong proof that SMA impairs the metabolic profile, and creatinine may be a key mediator. After this, showing how nusinersen therapy, which improves physical characteristics of patients, alters the metabolite profile. Presenting it this way, along with showing that globally nusinersen therapy does not alter the metabolite profile, the authors can show how individual metabolites important to SMA (i.e., creatinine) are altered by the therapy.
- There is a lot of discussion on inactivity models and SMA; however, a direct comparison between the groups was not examined in the current study despite this being in the title. Were the healthy controls inactive? Were measures of physical activity levels (accelerometers, questionnaires, etc.) examined between the two groups, as well as following nusinersen therapy? Is it possible to perform additional analysis with the current and the previous group’s work using the bed rest model (REF 37) to examine which metabolites are specific to SMA rather than inactivity?
- Please provide table for subject characteristics for all participants, and how physical function measures were altered by nusinersen therapy in SMA patients.
- Figures 1 and 2 are not referenced in the text, please add. Additionally, while the creatinine findings were a major conclusion of the study, this is not found in any of the figures.
- Can the authors elaborate why metabolites are overall lower in SMA group in comparison to healthy controls? Were any metabolites similar between the groups? If not, this could be a concern with potentially how the samples were collected between groups rather than group differences (i.e., artifact vs. physiological relevant).
Minor
- Unclear in abstract why nusinersen therapy is used in SMA patients, please add sentence why therapy is used in these patients.
How many metabolites were examined in the 1H-NM
Author Response
Response to Reviewer 2 Comments
The purpose of this study was to determine potential metabolites that relate to spinal muscular atrophy (SMA) and how nusinersen therapy alters the metabolite profile. To do this, 25 SMA patients (15 females, 10 males) were examined before and after 2-months of nusinersen therapy. Urine, serum, and liquor samples were collected for metabolome profiling by 1H-NMR. Urine samples were compared to 125 healthy controls (48 females, 77 males). Despite apparent phenotypic improvements, nusinersen therapy did not alter the metabolome profile in SMA patients. In comparison to healthy controls, SMA patients presented reduced number and concentration of metabolites, with machine learning confirming creatinine levels distinguish SMA patients from healthy controls. These findings suggest the improvements provoked by nusinersen therapy occur before, or devoid of changes in the metabolome profile. While the use of metabolome profiling has strength to identify molecular mediators of SMA, as well as potential mediators of nusinersen therapy, there was various issues with the overall manuscript to preclude it from being accepted in its current form. Comments/suggestions to improve the manuscript can be found below.
Major
R2.C1 The use of metabolome profiling has great strength to identify cellular mediators of diseased conditions. However, beyond the creatinine findings, no additional metabolites were discussed. Were other metabolites differentially expressed between the SMA and healthy controls?
A: In total, 674 chemical shifts present in ChenomX Library were considered in our analyses as described in our Materials and methods section.
We also described all the steps that lead us to present results that showed that in search of novel biomarkers at the levels of urine, liquor and serum that would separate between SMA before, SMA after and healthy cohorts only the creatinine exhibited reproducible signal to become eligible as a contributing variable.
We discussed the larger extent of variability in the SMA than in healthy cohort as the reason for such observation in the submitted manuscript.
We rephrased the text to make this point clear.
R2.C2 In the second to last paragraph in the Introduction, it appears that the authors specifically wanted to examine the role of creatinine with SMA, based on previous work. With a potentially large dataset, specifically looking at one metabolite minimizes the importance of looking at the whole metabolite profile. While creatinine may have been the metabolite that distinguished SMA the most, other metabolites may be involved and are worth presenting.
A: Please see also our disposition above.
We did consider all metabolites detected using 1H-NMR and did our best to analyse and screen for the potential biomarkers to the fullest possibilities.
Creatinine was the most important variable separating SMA from healthy males and females.
Other metabolites were also considered and analysed, however, did not prove of significant in separating the targeted groups.
We rephrased the text to make this point clear.
R2.C3 In addition to group differences, relationships between phenotypic characteristics can be used to identify potential cellular mediators of physical function measures (i.e., movement, posture, and strength tests described in the abstract). It would be beneficial to run a correlation matrix between metabolites and physical function measures to see which metabolites are associated with physical derangements with SMA, and which individual metabolites may be involved in physical improvements provoked by nusinersen therapy.
A: As presented in our submitted manuscript, there was no significant difference between the SMA before and after treatment neither in serum, urine or liquor matrices.
As there are no significant differences between the SMA before and SMA after groups, one has no statistical reason to start performing such correlative analyses as they would result in overfitted conclusions.
We rephrased the text for clarity.
R2.C4 Overall, a rearrangement of the findings will greatly improve the quality of the manuscript. Specifically, first presenting how SMA alters the metabolite profile in comparison to healthy controls will show strong proof that SMA impairs the metabolic profile, and creatinine may be a key mediator. After this, showing how nusinersen therapy, which improves physical characteristics of patients, alters the metabolite profile. Presenting it this way, along with showing that globally nusinersen therapy does not alter the metabolite profile, the authors can show how individual metabolites important to SMA (i.e., creatinine) are altered by the therapy.
A: Three distinct matrices (serum, liquor, urine) were used in search of metabolic differences between SMA before and after treatment. By doing this we first had to establish that urine can indeed be used to monitor SMA instead of serum and liquor that cannot ethically be collected from healthy participants.
After establishing this link, we could compare SMA to healthy urine samples, within which we reproducibly identified creatinine as responsible for separation between the two groups (SMA, healthy) and extended to other analyses reported in the manuscript.
Based on these reasons we have to disagree with this constructive comment.
We did our best to follow the comment by introducing several new figures: Figure2, Figure 3, Figure 5 and removed Figure 3 of our submitted manuscript. Electronic Supplementary Materials were amended with Figures S1 to S6.
Please see also our disposition to the comment above.
R2.C5 There is a lot of discussion on inactivity models and SMA; however, a direct comparison between the groups was not examined in the current study despite this being in the title. Were the healthy controls inactive? Were measures of physical activity levels (accelerometers, questionnaires, etc.) examined between the two groups, as well as following nusinersen therapy? Is it possible to perform additional analysis with the current and the previous group’s work using the bed rest model (REF 37) to examine which metabolites are specific to SMA rather than inactivity?
A: We modified and simplified the title as suggested by Reviewer1 that also partially addresses this comment. No, the controls were healthy normally active males and females of matching age.
Questionnaires targeting SMA and healthy groups of males and females in this study showed that no significant difference existed in the extent of physical activity and lifestyle of SMA patients before and after SMA therapy, whereas significant differences existed in the extent of physical exercise between the SMA and healthy matched controls, exhibiting average active population lifestyle (posture, daily exercise, work, school, everyday activities).
We rephrased the manuscript to make this information explicitly clear in Materials and methods section where participants were described in the submitted manuscript.
In addition, a novel Table1 was prepared where additional information was presented in structured was in the revised manuscript.
R2.C6 Please provide table for subject characteristics for all participants, and how physical function measures were altered by nusinersen therapy in SMA patients.
A: A novel Table1 was prepared as requested.
R2.C7 Figures 1 and 2 are not referenced in the text, please add. Additionally, while the creatinine findings were a major conclusion of the study, this is not found in any of the figures.
A: Thank you. Corrected.
There are now three novel figures, in total five that are part of the revised manuscript, next to six figures in the revised Electronic Supplementary Material. Figure 3 of the submitted manuscript was removed as suggested above.
R2.C8 Can the authors elaborate why metabolites are overall lower in SMA group in comparison to healthy controls? Were any metabolites similar between the groups? If not, this could be a concern with potentially how the samples were collected between groups rather than group differences (i.e., artifact vs. physiological relevant).
A: Additional explanation and references were provided in the revised manuscript.
Minor
R2.C9 Unclear in abstract why nusinersen therapy is used in SMA patients, please add sentence why therapy is used in these patients.
A: We added “Nusinersen intrathecal application was the first therapy for treatment of SMA by the FDA and EMA”.
R2.C10 How many metabolites were examined in the 1H-NMR
A: As described in our initially submitted manuscript:
This information is already present at the end of the Materials and methods section of the submitted manuscript. In addition, we revised our Materials and methods section and explicitly provided information that using ChenomX chemical shift library 674 metabolic chemical shifts were used in screening of metabolites.

Round 2
Reviewer 1 Report
General comment:
Deutsch and colleagues have thoroughly adapted their manuscript according to the suggestions during the first round of revisions and are complemented for their comprehensive work. This reviewer believes that in the process, the manuscript has become substantially more insightful to the reader and improved its overall quality. Pending the changes to the manuscript outlined below, the publication of the manuscript in “metabolites” will be endorsed by this reviewer.
Minor revisions:
Line 151: Correct the table column “Oter” to “Other”.
Line 179-182: Consider amending to “the relative ease [...], however, supports the selection of 1H-NMR [...].”.
Line 194-198: Do these statistical values here refer to the sum of all analyzed metabolites?
Line 230: Consider deleting “recent” when referring to the Nieman et al. study from 2013.
Line 250-253: Please specify what the numbers on the scale display (units) and what type of scale it is (i.e. log-scale, fold-change or effect size).
Line 255-258: See comment above. Furthermore, the heatmaps are currently suggesting very robust changes. Not only between sexes (figure 3b), but also between healthy and SMA (figure 2a-b). In that context, it is surprising that creatinine would be the only metabolite being significantly different (as outlined in line 272-276). Consider a milder color scheme for both heatmaps (figure 2 and 3), that more closely reflects the fact that the group differences were small. Also consider plotting the patients (and healthy controls) as individual squares, to represent the variability in the data more accurately.
Line 272-276: Clarify for the reader that it is a decrease in creatinine, which distinguishes SMA from healthy control in your study. With the big role that the identification of creatinine as biomarker in SMA plays in this manuscript, the nature of the difference between the groups should be stated more directly.
Line 295: Given the great differences between control and SMA patients, consider using a log scale to display the creatinine data. Especially since in line 203-207 the authors mention that sex effects were also apparent for SMA. This is currently not represented by how the data is displayed in figure 5.
Line 299-300: Please list or mention the fact that SMA is primarily a disease affecting motor neurons, which in turn has diverse consequences for multiple tissues. The current phrasing appears as though SMA would indiscriminately affect organs.
Author Response
Response to Reviewer 1 Comments
Deutsch and colleagues have thoroughly adapted their manuscript according to the suggestions during the first round of revisions and are complemented for their comprehensive work. This reviewer believes that in the process, the manuscript has become substantially more insightful to the reader and improved its overall quality. Pending the changes to the manuscript outlined below, the publication of the manuscript in “metabolites” will be endorsed by this reviewer.
Line 151: Correct the table column “Oter” to “Other”.
A: “Oter” was changed to “Other”.
Line 179-182: Consider amending to “the relative ease [...], however, supports the selection of 1H-NMR [...].”.
A: This sentence was amended.
Line 194-198: Do these statistical values here refer to the sum of all analyzed metabolites?
A: Statistical values presented in this part refer to data matrices containing all analyzed metabolites individually for all samples and not the sum of all analyzed metabolites.
Line 230: Consider deleting “recent” when referring to the Nieman et al. study from 2013.
A: “recent” was deleted.
Line 250-253: Please specify what the numbers on the scale display (units) and what type of scale it is (i.e. log-scale, fold-change or effect size).
A: Heatmaps were generated according to PLSDA variable importance in projection (VIP) scores. Sentences explaining this part was added in the text under both figures.
Line 255-258: See comment above. Furthermore, the heatmaps are currently suggesting very robust changes. Not only between sexes (figure 3b), but also between healthy and SMA (figure 2a-b). In that context, it is surprising that creatinine would be the only metabolite being significantly different (as outlined in line 272-276). Consider a milder color scheme for both heatmaps (figure 2 and 3), that more closely reflects the fact that the group differences were small. Also consider plotting the patients (and healthy controls) as individual squares, to represent the variability in the data more accurately.
A: Heatmaps representing individual participant metabolomes were included in the ESM as supplementary Figure S7 and Figure S8.
Line 272-276: Clarify for the reader that it is a decrease in creatinine, which distinguishes SMA from healthy control in your study. With the big role that the identification of creatinine as biomarker in SMA plays in this manuscript, the nature of the difference between the groups should be stated more directly.
A: This sentence was amended to clarify decreasing creatinine concentrations.
Line 295: Given the great differences between control and SMA patients, consider using a log scale to display the creatinine data. Especially since in line 203-207 the authors mention that sex effects were also apparent for SMA. This is currently not represented by how the data is displayed in figure 5.
A: We changed the y-axis with log transformation and moved original figure to ESM (Figure S9).
The differences exist in urine metabolomes between females and males, but these differences were significant in healthy group (p<0.05) and were less pronounced in comparing SMA females and males (p<0.85).
Line 299-300: Please list or mention the fact that SMA is primarily a disease affecting motor neurons, which in turn has diverse consequences for multiple tissues. The current phrasing appears as though SMA would indiscriminately affect organs.
A: We rephrased this sentence as requested.

Reviewer 2 Report
The quality of the manuscript has greatly improved with the changes. Minor changes are recommended below.
Minor Comments
- Add scale for heatmaps on new figures.
- Heatmaps appear to be average of the groups, please include values for all participants for each group in the heatmaps.
Author Response
Response to Reviewer 2 Comments
The quality of the manuscript has greatly improved with the changes. Minor changes are recommended below.
- Add scale for heatmaps on new figures.
A: Heatmaps were generated according to PLSDA variable importance in projection (VIP) scores. Sentences explaining this part was added in the text under both figures.
- Heatmaps appear to be average of the groups, please include values for all participants for each group in the heatmaps.
A: Heatmaps represented individual participant metabolomes were included in the ESM (Figures S7 and S8).
